# The LUMirage: An independent evaluation of zero-shot performance in the LUMIR challenge

**Rohit Jena**                                              RJENA@SEAS.UPENN.EDU
**Pratik Chaudhari**                                PRATIKAC@SEAS.UPENN.EDU
**James Gee**                                                  GEE@UPENN.EDU
*University of Pennsylvania*

## Abstract

The LUMIR challenge represents an important benchmark for evaluating deformable image registration methods on large-scale neuroimaging data. While the challenge demonstrates that modern deep learning methods achieve competitive accuracy on T1-weighted MRI, it also claims exceptional zero-shot generalization to unseen contrasts and resolutions—assertions that contradict established understanding of domain shift in deep learning. In this paper, we perform an independent re-evaluation of these zero-shot claims using rigorous evaluation protocols while addressing potential sources of instrumentation bias. Our findings reveal a more nuanced picture: (1) deep learning methods perform comparably to iterative optimization on in-distribution T1w images and even on human-adjacent species (macaque), demonstrating improved task understanding; (2) however, performance degrades significantly on out-of-distribution contrasts (T2, T2*, FLAIR), with Cohen's d scores ranging from 0.7–1.5, indicating substantial practical impact on downstream clinical workflows; (3) deep learning methods face scalability limitations on high-resolution data, failing to run on 0.6mm isotropic images, while iterative methods benefit from increased resolution; and (4) deep methods exhibit high sensitivity to preprocessing choices. These results align with the well-established literature on domain shift and suggest that claims of universal zero-shot superiority require careful scrutiny. We advocate for evaluation protocols that reflect practical clinical and research workflows rather than conditions that may inadvertently favor particular method classes.

**Keywords:** image registration, deep learning, instrumentation bias, neuroimaging, foundational models

## 1. Introduction

Quantitative analysis and integration of biomedical and biological data requires images to reside in a common coordinate frame. Toward this end, deformable image registration (DIR) is a key workhorse operation in medical image analysis, enabling analysis and fusion of data into a common coordinate frame. Deformable Image Registration is an inverse task and shares the classical problems shared by most inverse problems in computer vision - ill-posedness, susceptibility to noise and artifacts, non convexity, and a lack of well-defined ground truth solutions. Most successful registration algorithms use a variational approach to find the optimal transformation using iterative optimization, subject to constraints and regularizations on the transformation. Although these methods are very robust to a wide range of modalities, anatomies, and species, early approaches were typically implemented on CPU, making them prohibitively slow for large scale studies. Recent methods have addressed this limitation by proposing very fast GPU implementations (Jena et al., 2024a;

Mang, 2024; Siebert et al., 2024) that employ advanced optimizers and efficient implementations, often performing iterative optimization in under a second for clinical volumes, and scale very efficiently to large-scale problems for bespoke applications in life sciences (Wang et al., 2020; Kronman et al., 2023). Deep learning-based methods for image registration take a fundamentally different approach by posing the inverse problem as a statistical learning problem and using feedforward inference to substitute hundreds of iterations of an iterative optimizer with a few (tens to hundreds of) layers to predict a deformation field directly. While deep learning methods can substantially benefit by sidestepping iterative optimization, and learning to explicitly register anatomical ROIs from auxiliary information such as labelmaps or landmarks, most deep methods suffer from generalization to out-of-domain contrasts, and resolutions. This is a typical property of most parametric modelling encompassing all of statistical learning theory - the train and test data is assumed to be from the same distribution (Hastie, 2009; Vapnik, 1998). To mitigate the distribution shift issue, methods like SynthMorph (Hoffmann et al., 2021) propose training on a combinatorial space of synthetically generated volumes. Other foundational models (Tian et al., 2024) propose training on a wide range of contrasts and anatomies to learn a general purpose registration network. Improving robustness of deep networks to work on a long tail of unseen modalities is an area of active research.

Despite the saliency and centrality of the registration problem across many workflows in biomedical imaging, existing evaluation challenges have been limited in scope compared to other tasks in medical imaging like segmentation for providing insights into the benefits and limitations of approaches in registration in deep learning. The LUMIR challenge (Chen et al., 2025) aims to address these limitations with a large scale dataset and a platform for benchmarking and advancing the next generation of registration algorithms with the goal of advancing clinical workflows and neuroscience research. The challenge evaluation shows that modern deep learning methods can achieve competitive accuracy and high inference efficiency when trained on millions of T1w MRI image pairs without additional labeled supervision. This is a highly encouraging result, showing the maturity of deep learning methods on large scale neuroimaging datasets. However, the paper makes a few more claims on zero-shot performance of deep networks that defy commonsense knowledge about how parametric statistical models work:

- **Training deep learning models for registration on T1w MRI brain images alone performs exceptionally well on unseen resolutions and contrasts, even outperforming methods that are specifically trained to be domain-agnostic**: This is a claim founded neither in theory nor in practice. It is almost universally known that deep networks suffer on out-of-distribution data (i.e. deep learning methods are not good at extrapolation), that has led to numerous contributions in domain adaptation, transfer learning, self-supervised learning, and synthetically generated datasets and environments. It is possible that a special design proposed in a registration network can disentangle the modality completely and possibly lead to good domain-agnostic performance. But in that case, only few such explicit designs should perform well on out-of-distribution images. However, the paper claims that *all deep methods* outperform iterative methods on all out-of-distribution modalities. We find this claim not very plausible without additional theoretical or empirical justification, and therefore test this claim independently in the paper.

- **Deep learning methods are unequivocally superior to iterative optimization methods on out-of-distribution T1w images**: This claim may be plausible since newer deep learning methods use registration-specific designs that are inspired by components used in iterative optimization methods. Specific design decisions may contribute to robust and accurate performance, but the free lunch theorem suggests that a universally superior optimization technique on T1w images is unlikely.

In this paper, we perform a systematic and thorough re-evaluation of the claims about zero-shot performance made in the LUMIR challenge evaluation. Our conclusions are rather unsurprising, but strikingly different than in Chen et al. (2025). *First*, we observe that the performance of SOTA deep learning methods on T1 weighted MRI imaging is indeed comparable with iterative optimization methods, even on a human-adjacent species like Macaque - showing that the next generation of deep learning algorithms for registration demonstrate substantially better task understanding. However, deep learning methods can show inferior performance on highly parcellated regions like the SLANT segmentation, compared to other segmentation labels like DeepAtropos or SynthSeg. *Second*, the task understanding does not translate to better performance on out-of-distribution contrasts, contrary to the results shown in Chen et al. (2025). *Third*, scalability remains an issue with deep learning methods as demonstrated on the high-resolution Ultracortex dataset, while iterative optimization methods enjoy improved performance by registering high resolution brains due to their low memory footprint. The scalability makes iterative optimization method a more practical choice for high-resolution image registration pertinent in histopathological workflows and life sciences research. *Fourth*, we show that deep learning methods are highly sensitive to even trivial changes in image preprocessing, including retaining padding from the original dataset. These modes of sensitivity puts a burden on the practitioner to ensure the data conforms to the emulated preprocessing standards during training, potentially limiting its applications to high variability pertinent in real clinical scenarios.

## 2. Evaluation Setup

The LUMIR challenge shows zero-shot evaluation on a variety of datasets spanning different contrasts, two species, and three tasks (inter-subject, atlas-to-subject, and subject-to-atlas registration). However, the labeled data generation and evaluation are not discussed in sufficient detail to ensure reproducibility. There are also few oversights in the dataset descriptions and evaluation criteria that we discuss, and consider their effect in our evaluation. For each dataset, we also consider the primary sources of instrumentation bias that can affect evaluation, and how we control for these conditions.

### 2.1. Primary Sources of Instrumentation Bias

To ensure that evaluation is fair, we discuss common and data-specific sources of instrumentation biases that can affect evaluation. Acknowledging instrumentation bias is important because the evaluation in challenge datasets may be significantly different than how a practitioner uses the methods in clinical and research workflows. Our previous work (Jena et al., 2024b) shows that deep learning methods typically exhibit instrumentation bias that lead

to misrepresentation of the true performance of optimization methods. Primary sources of instrumentation bias include:

- **Running multimodal registration with unimodal similarity functions**: Iterative solvers will catastrophically fail if multimodal images are attempted to be registered using unimodal losses. For example, the Ultracortex dataset contains a mix of MP-RAGE and MP2RAGE sequences for different subjects, which are qualitatively and quantitatively distinct in terms of contrast and resolution. Our evaluation for iterative optimization considers the effect of choosing different similarity functions for multimodal registration.

- **Evaluating registration algorithms on low resolution images**: Most modern registration challenges downsample the data into a standard isotropic resolution and attempt to fit their training data, due to high memory requirements. Two major benefits of using iterative optimization methods is their very low memory footprint at inference, and that they perform *better* at high resolutions. In almost every practical scenario, a practitioner would desire the images to be registered at the highest resolution possible to obtain high fidelity warp fields. Running optimization solvers on downsampled resolutions therefore constitutes *intentional weakening* of the baseline. Instead, DLIR proponents must focus on improving the capabilities of deep learning to register large scale images, rather than weakening optimization solvers.

- **Labelmap bias due to non-existent intensity boundaries**: SLANT is used for labelling, which uses the BrainCOLOR protocol to obtain a comprehensive segmentation of the brain. However, cortical parcellation is performed by lifting the atlas to the subject coordinate frame; the cortical boundaries do not exist as intensity features in the in-vivo MRI. This leads to spurious results when Dice score is computed (Jena et al., 2024b). For in-vivo intensity images, cortical and subcortical structures that *can* be delineated must be included in evaluation.

The LUMIR challenge claims to perform evaluation on a variety of resolutions, but the text mentions that all datasets are resampled to the 1mm MNI space, essentially discarding the effect on performance due to the varying resolution of the datasets. Moreover, we find that registering the PRIME-DE dataset to the MNI template is somewhat questionable, and our evaluation leads to a smaller discrepancy in performance between deep and iterative methods by improving the performance of the deep learning method.

### 2.2. Choice of Baselines

Chen et al. (2025) predicate that the new generation of deep learning architectures surpass optimization solvers on all registration tasks. To carefully evaluate this claim, we consider independently evaluating the top performing methods ranked in Table 1 of the LUMIR challenge paper, with FireANTs (Jena et al., 2024a) which is reported as the best performing iterative solver. However, at the time of writing, out of the top eight best performing methods, only *two* implementations are available in the public domain: SITReg (Honkamaa and Marttinen, 2023) (Rank 1) and Vector Field Attention (Rank 4) (Liu et al., 2024). Despite SITReg providing an open-source implementation, it does not provide user friendly interfaces for evaluation on arbitrary datasets and despite our best efforts at modifying

the codebase, we could not run the trained model on our evaluation setup. VFA on the other hand provided highly customizable configurations that allowed us to seamlessly run evaluations with minimal changes to the original codebase. FireANTs (Rank 12) provides both CLI-based and Python-based scripts for evaluation of arbitrary datasets, and we use the Python-based script for consistency. Therefore, we use VFA as the primary deep learning method for comparison with FireANTs.

## 3. Inter-subject registration on the NIMH T1w dataset

The National Institute of Mental Health (NIMH) Data Archive uses human subject data collected from hundreds of research projects across many scientific domains. We use the Research Volunteer Dataset that characterizes healthy adult research volunteers in clinical assessments using mood-related psychometrics, cognitive function neurophysiological tests, structural and functional MRI, DRI, and MEG. We use a subset of the T1w MRI dataset for inter-subject registration.

T1w MRI provides excellent gray/white matter contrast, and is routinely used for structural segmentation, and morphometry. Similar to most methods, we use overlap of cortical and subcortical structures for evaluation. The LUMIR challenge's evaluation protocol uses a tool called SLANT (Huo et al., 2019) that uses a deep learning model to segment a T1 MRI scan into 133 labels based on the BrainCOLOR protocol (Klein et al., 2010). However, there are a few issues with using SLANT for evaluation. *First*, SLANT produces 133 labels, but is trained only on 45 T1-weighted MRI scans from the OASIS dataset. This can lead to a significant lack of generalization to other modalities like T2w, T2*, FLAIR, and Ultra High Field (UHF) MRI. While label fusion can help, systematic bias in the multi-atlas fusion step can propagate into the learned UNet. We observe degradation in performance of the SLANT algorithm on the Ultracortex dataset. *Second*, measures like Dice Scores are highly sensitive to the volume of the structures, and consequently the choice of interpolation method can significantly affect the score. For example, in our experiments, changing the interpolation method from trilinear to nearest neighbor in VFA leads to a drop of about 10 points in Dice score for the SLANT labelmaps. To ensure fair comparison, we fix the label interpolation scheme wherein we first convert each labelmap to a binary mask, perform trilinear interpolation to obtain probability maps for each label, and for each voxel select the label with the highest probability. This interpolation scheme avoids blocky artifacts introduced by nearest neighbor interpolation, considers partial volume effects of the probability maps, and assigns a single label to each voxel. *Third*, our previous work (Jena et al., 2024b) shows that the mutual information between images and label maps is correlated with Dice Score of registration. The BrainCOLOR protocol used in SLANT provides extensive fine grained structures including sulcal/gyri boundaries, and various lobe boundaries, which cannot be delineated by intensity features alone. This can lead to Dice scores of registration methods capturing spurious associations rather than anatomical relationships that can be delineated by intensity features since we are interested in evaluating intensity-based registration methods.

**Evaluation**. To address all these issues, we choose three labelling protocols with varying degrees of granularity and anatomical coverage: (1) SLANT as used in the original LUMIR challenge, (2) SynthSeg (Billot et al., 2023) for a comprehensive segmentation

of various subcortical structures while segmenting the cerebral cortex as a single label for each hemisphere, and (3) DeepAtropos (Tustison et al., 2021) that provides a coarse six label segmentation of CSF, GM, WM, deep GM, brainstem, and cerebellum. We randomly choose 100 subjects from the dataset, resample to 1mm isotropic resolution, and apply all three segmentation protocols to obtain labelmaps. This provides us a total of 9900 image pairs for evaluation. To provide robust estimates, we crop the bottom five percentile of the Dice scores for each registration method. We provide common statistical measures (mean, median, standard deviation) for the Dice scores of the three registration methods in Table 1, and violin plots in Figure 1.

**Significance Tests.** To evaluate the practical impact of the differences in labelling protocols, we perform a paired t-test and a Wilcoxon signed rank test between the Dice scores of the three registration methods. For such a high sample size, statistical significance ($p < 0.05$) is almost guaranteed for any difference, and we observed p-values lower than $10^{-4}$ for all method pairs. To report statistical significance, we take inspiration from Klein et al. (2009) and perform permutation tests (Menke and Martinez, 2004) to determine if the means of a small set of independent overlap values obtained by each of the registration methods are the same. The subset of brain pairs was selected so that each brain was used only once, and we fixed the number of permutations to 1024, and calculate 10,000 p-values for each method pair. We report the fraction of p-values less than 0.05 (represented as $\mu$) for each method pair as a proxy for statistical significance, as suggested by Klein et al. (2009), with higher values indicating greater statistical significance. To measure practical impact, we measure Cohen's d (that represents effect sizes) for practical significance ($d > 0.2$) for each pair of registration methods.

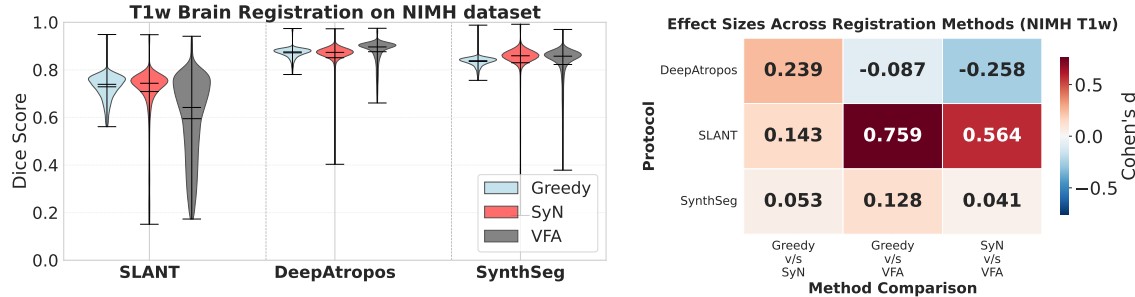

Figure 1: **Comparison of the three registration methods on the NIMH T1w dataset. Left** shows violin plots of the Dice scores of the top iterative and deep learning registration methods on the NIMH T1w dataset. **Right** shows Cohen's d scores for all method pairs, quantifying the practical significance of the differences in Dice scores between the three registration methods.

**Results.** Interestingly, VFA performs significantly worse on the SLANT labelmaps (Table 1), in direct contrast to the results in the LUMIR challenge. Since the conditions for evaluation of the original challenge are unspecified, we can only speculate that the differences are due to preprocessing conditions and label interpolation schemes. On the SynthSeg and DeepAtropos labelmaps, the performance of VFA is comparable to (but still lower than)

| Method | SLANT | | | DeepAtropos | | | SynthSeg | | |
|--------|-------|--------|--------|-------------|--------|--------|----------|--------|--------|
|        | Mean  | Median | Std    | Mean        | Median | Std    | Mean     | Median | Std    |
| Greedy | 0.7289 | 0.7393 | 0.0547 | 0.8717 | 0.8755 | 0.0219 | 0.8356 | 0.8384 | 0.0232 |
| SyN    | 0.7090 | 0.7437 | 0.1178 | 0.8511 | 0.8735 | 0.0844 | 0.8300 | 0.8593 | 0.1183 |
| VFA    | 0.5950 | 0.6421 | 0.1700 | 0.8764 | 0.8964 | 0.0507 | 0.8227 | 0.8575 | 0.0933 |

Table 1: **Registration method performance across different labelling protocols on the NIMH T1w dataset.** Table shows the mean, median, and standard deviation of the Dice scores of the top three registration methods on the NIMH T1w dataset.

Greedy and SyN, with minor differences in the Cohen's d scores (Figure 1). Permutation tests in Table 4 show that VFA significantly underperforms Greedy and SyN on the SLANT labelmaps indicated by high $\mu$ values, while lower $\mu$ values for DeepAtropos and SynthSeg labels suggest that the differences in Dice scores for these labelmaps are not statistically significant. This is an indicator that modern deep methods like VFA are able to register coarse anatomical structures well, but may struggle with highly parcellated structures. However, deep methods are comparable to iterative methods on inter-subject registration of in-distribution contrast, showing maturity of deep learning methods in terms of task understanding for image registration, compared to the previous generation of methods that performed well on the training data but failed to generalize to modest and practical amounts of domain shift (Jena et al., 2024b; Jian et al., 2024; Jena et al., 2025; Jian et al., 2025; Liu et al., 2025).

## 4. Inter-subject registration on the PRIME-DE Macaque dataset

A natural extension of zero-shot evaluation from the T1w human MRI is to evaluate registration performance on a human-adjacent mammalian species like the Macaque. To that end, the PRIME-DE dataset provides a collection of T1w MRI images of the Macaque brain, with original resolutions varying from 0.3 to 0.8mm. This is in contrast to Chen et al. (2025) which incorrectly claims that the brain images are originally acquired at 1mm isotropic resolution, indicating a potential lack of quality control in the original evaluation. We download data from the five different sites mentioned in the original challenge, followed by brain extraction and segmentation using the nBEST (Zhong et al., 2024) tool. All subjects are affinely registered using FireANTs to a manually chosen subject with 0.3mm resolution, to maximize the field of view and resolution for subjects with lower resolution. We obtain 116 brain images, resulting in 13,340 (= 116 × 115) image pairs for evaluation. We include preprocessing and affine alignment scripts in our provided code. The nBEST tool provides two segmentations: (1) segmentation of three cortical labels (GM, WM, CSF) and (2) segmentation of six subcortical labels, including thalamus, caudate, putamen, pallidum, hippocampus, and amygdala.

**Results**. We include violin plots, summary statistics, and Cohen's d scores of Dice score overlap between inter-subject registered labelmaps for both the cortical and subcortical segmentations in Figure 2 and Figure 3. We note a small gap between the performance of Greedy and VFA for both cortical and subcortical segmentations, showing that modern

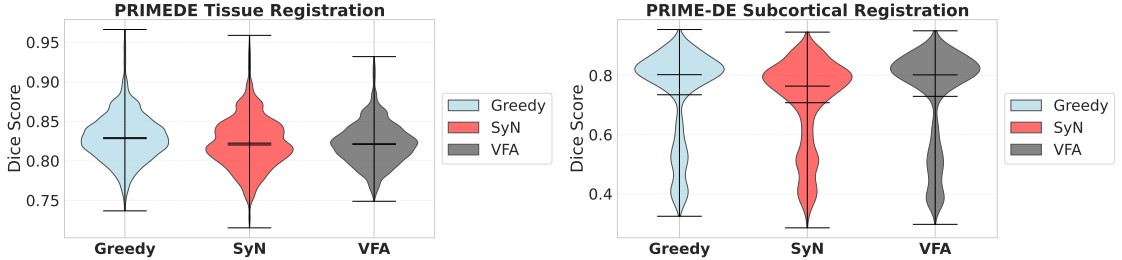

Figure 2: **Comparison of the three registration methods on the PRIME-DE dataset.**
**Left** shows violin plots of the Dice scores of tissue overlap (GM, WM, CSF), **Right**
shows violin plots of the Dice scores of subcortical overlap between the registered and
reference labelmaps.

| Method | Tissue | Subcortical |
|--------|--------|-------------|
| Greedy | $0.829 \pm 0.030$ | $0.735 \pm 0.158$ |
| SyN | $0.823 \pm 0.032$ | $0.708 \pm 0.151$ |
| VFA | $0.822 \pm 0.026$ | $0.729 \pm 0.165$ |

**Effect Sizes Across Registration Methods (PRIME-DE)**

| Protocol | Greedy v/s SyN | Greedy v/s VFA | SyN v/s VFA |
|----------|------|------|------|
| Subcortical | 0.778 | 0.308 | -0.530 |
| Tissue | 0.971 | 1.016 | 0.069 |

**Method Comparison**

Figure 3: **Quantitative comparison of the three registration methods on the PRIME-
DE dataset. Left** shows the mean, median, and standard deviation of the Dice scores
of the top three registration methods on the PRIME-DE dataset. **Right** shows Cohen's
d scores for all method pairs.

deep learning methods demonstrate improved task understanding on a familiar modality
but unseen anatomy (i.e. T1w MRI of the macaque cerebrum). The Cohen's d scores in
Figure 3 show that although small, the performance difference between Greedy and VFA
is of practical significance for both cortical and subcortical segmentations, with d scores of
0.308 and 1.016, significantly outside the accepted standard for "small effects" ($d < 0.2$).
Permutation tests in Table 2 also indicate that the difference in Dice scores between Greedy
and VFA *are* statistically significant for independent subsets of image pairs for cortical
segmentations, and slightly less but still significant for subcortical segmentations. However,
SyN underperforms VFA *significantly* for the subcortical segmentations, indicating that
Greedy is a better overall choice for iterative registration. This modest performance gap is
in contrast to the results in Chen et al. (2025) where VFA underperformed FireANTsGreedy
substantially, which could be attributed to poorly designed preprocessing conditions in the
original evaluation. This underscores the importance of careful design of preprocessing
conditions for zero-shot evaluation, and the need for a standardized evaluation protocol for
inter-subject registration.

| Method | Cortical | Subcortical |
|---|---|---|
| Greedy v.s. SyN | 1.0 | 1.0 |
| Greedy v.s. VFA | 1.0 | 0.6873 |
| SyN v.s. VFA | 0.0512 | 0.9990 |

Table 2: Statistical significance on the PRIME-DE dataset represented as fraction of p-values less than 0.05 for each method pair using permutation tests. Higher values represent greater statistical significance.

## 5. Out-of-distribution contrasts on the NIMH T1w dataset

While T1-weighted imaging provides excellent anatomical detail with superior gray-white matter contrast ideal for morphometric analysis and structural segmentation, it exhibits limited sensitivity to many pathological processes. T2-weighted and FLAIR sequences offer complementary contrast mechanisms that are essential for detecting white matter lesions, edema, inflammation, and demyelination—pathologies that often have subtler appearance on T1w scans. For example, T2* imaging is sensitive to magnetic susceptibility effects, making it useful for detecting hemorrhages, iron buildup, and other magnetic substances that are not visible in T1w scans. T2 weighted images are particularly useful for visualizing fluid-filled structures, such as cerebrospinal fluid (CSF) and white matter lesions that appear isointense with the background on T1w scans. FLAIR images on the other hand suppresses signal from CSF, highlighting abnormalities like lesions, tumors, and stroke against a more homogeneous background. In these modalities, the GM-WM boundary is often less distinct than in a T1w scan. In clinical practice, multimodal protocols combining these contrasts are standard precisely because no single sequence provides comprehensive tissue characterization: T1w reveals anatomy while T2/FLAIR/T2* reveal pathophysiology.

The LUMIR challenge uses the NIMH dataset with T1w, T2w, T2*, and FLAIR sequences for zero-shot evaluation of out-of-distribution contrasts, where they use the SLANT segmentation from the co-registered T1w images to the T2w, T2*, and FLAIR images to obtain labelmaps. However, since the majority of labels in the SLANT segmentation are cortical parcellations and the T2w, T2*, and FLAIR images do not provide sufficient GM-WM contrast compared to T1w images, we argue that this parcellation is not representative of the registration task. Moreover, our experiments in Section 3 show that the performance of deep learning methods (VFA) on the SLANT labelmaps is significantly worse than iterative methods (Greedy and SyN). Therefore, we consider SynthSeg for labelmap generation on the T2w, T2*, and FLAIR images. SynthSeg is a general purpose segmentation model that is trained on a wide range of contrasts, and is suitable for accurate segmetnation for all three contrasts. Moreover, VFA performs comparably to iterative methods on the SynthSeg labelmap on T1w images, setting a benchmark for performance comparison between in-distribution and out-of-distribution contrasts. Initially, we segmented 438 images from each contrast, leading to a total of 191,406 (= 438 × 437) image pairs for evaluation of each contrast. We evaluate the average Dice Score overlap between the registered and reference

labelmaps on a randomly chosen (and fixed) subset of 5,000 pairs for each contrast to reduce computational cost.

**Results**. Violin plots and pairwise Cohen's d scores are reported in Figure 4 and statistical summaries are shown in Table 3. Compared to T1w images, the performance of the top deep learning method VFA drops significantly for unseen contrasts. The largest difference in performance is observed in T2w images, followed by T2* and FLAIR images. The Cohen's d scores in Figure 4 underscore that the practical impact of the performance difference is significant for all three contrasts, contrary to the results in the T1w dataset where the differences are minor and do not have significant practical impact. For example, on the T2 and T2* images, the Cohen's d scores are in the range of 0.70-1.51, significantly outside the accepted standard for "small effects" ($d < 0.2$). Permutation tests (Table 4) further confirm that these performance gaps are statistically significant. The consistently large mean test statistics ($\mu$) for T2 and T2* reinforce that the observed differences reflect systematic performance degradation rather than sampling variability. In contrast, both Cohen's d and permutation test results indicate smaller effect sizes and weaker statistical significance for FLAIR. This aligns with the long-tailed Dice distribution observed in the violin plots (Figure 4), suggesting higher variability but less consistent separation between methods. We hypothesize that this behavior stems from the contrast properties of FLAIR imaging: FLAIR emphasizes T2-hyperintense pathology (e.g., white matter lesions, edema, periventricular abnormalities) while providing comparatively weak contrast at morphometric tissue boundaries such as the GM-WM interface and deep subcortical structures. The resulting boundary ambiguity likely increases registration variability of morphometric boundaries without producing a consistent directional performance gap between methods.

These results are in stark contrast to the results in the LUMIR challenge, where VFA performed *better* than iterative methods on out-of-distribution contrasts, with seemingly no empirical consideration or theoretical justification for the observed difference. Domain shift is an established problem in deep learning, and is a well-studied phenomena that is pervasive in a broad range of application areas, and has garnered significant resources to systematically study and mitigate it (Beery et al., 2020; Zech et al., 2018; AlBadawy et al., 2018; Jadon et al., 2025). The LUMIR challenge asserts that a variety of deep learning architectures are robust to domain shift just by training on a large set of T1w images, a direct contradiction to the extensive literature on domain shift and domain adaptation in deep learning. A crucial question therefore emerges: can this seemingly absurd conclusion from the original challenge be extended to other tasks like lung CT or abdomen registration? Moreover, the LUMIR challenge did not explicitly discuss or account for the interaction between registration performance and image contrast. In particular, T1w and FLAIR images emphasize qualitatively different structures—T1w highlights morphometric boundaries, whereas FLAIR emphasizes T2-hyperintense pathology. Because these contrasts are complementary rather than equivalent, registration difficulty and evaluation metrics may reflect modality-specific contrast properties rather than purely methodological differences. Our results are consistent with the expectation that deep methods learn a distribution of *T1w (task) specific* features that is then used in conjunction with registration-aware modules (Jian et al., 2025, 2024) to achieve generalization to T1w images. Moreover, VFA exhibits significantly higher variance (a proxy for predictive variance) than Greedy and SyN for all three contrasts, which is consistent with the literature on predictive uncertainty and entropy

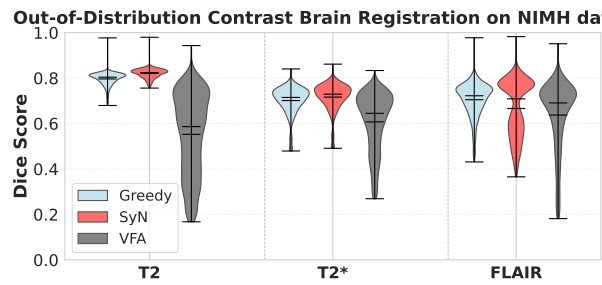
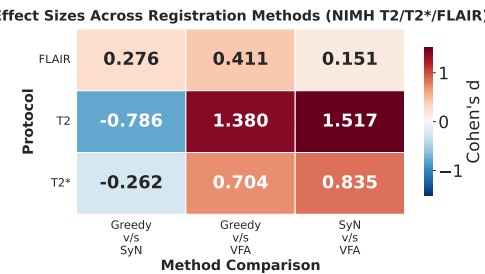

Figure 4: Comparison of the three registration methods on out-of-distribution contrasts on the NIMH dataset with labels generated by SynthSeg.

estimation of deep learning methods on out-of-distribution data (Lakshminarayanan et al., 2017; Maddox et al., 2019; Malinin and Gales, 2018).

| Method | T2 | | | T2* | | | FLAIR | | |
|--------|------|--------|-----|------|--------|-----|-------|--------|-----|
| | Mean | Median | Std | Mean | Median | Std | Mean | Median | Std |
| Greedy | 0.7961 | 0.8038 | 0.0292 | 0.7012 | 0.7148 | 0.0608 | 0.7049 | 0.7222 | 0.0732 |
| SyN | 0.8203 | 0.8241 | 0.0213 | 0.7161 | 0.7292 | 0.0606 | 0.6662 | 0.7087 | 0.1245 |
| VFA | 0.5524 | 0.5865 | 0.1792 | 0.6072 | 0.6450 | 0.1287 | 0.6373 | 0.6907 | 0.1509 |

Table 3: Registration method performance across different out-of-distribution contrasts on the NIMH dataset with labels generated by SynthSeg.

## 6. Inter-subject registration on the Ultracortex dataset

The Ultracortex dataset (Mahler et al., 2024) includes a collection of 9.4T ultra-high field MRI images of the human brain, with resolutions varying from 0.6 to 0.8mm. The images are acquired using a 9.4T MRI scanner, and the data consists of both MP-RAGE and MP2RAGE sequences. The dataset includes high-quality manual segmentations for 12 subjects - which include both gray and white matter segmentations for each hemisphere - leading to 4 labels.

| Method | SLANT | DeepAtropos | SynthSeg | | | |
|--------|-------|-------------|------|------|------|-------|
| | T1 | T1 | T1 | T2 | T2* | FLAIR |
| Greedy v.s. SyN | 0.1870 | 0.6383 | 0.0031 | 0.5930 | 0.4292 | 0.0443 |
| Greedy v.s. VFA | 1.0000 | 0.0222 | 0.1258 | 0.4244 | 0.2407 | 0.0280 |
| SyN v.s. VFA | 0.9781 | 0.2143 | 0.0025 | 0.6035 | 0.4629 | 0.0292 |

Table 4: Statistical significance on the NIMH dataset represented as fraction of p-values less than 0.05 for each method pair using permutation tests. Higher values represent greater statistical significance.

The LUMIR challenge uses SLANT to obtain labelmaps for the Ultracortex dataset, and downsamples the images to 1mm isotropic. However, this preprocessing has two undesirable effects. *First*, submillimeter resolution images can provide additional cytoarchitectural detail and act as a bridge between low resolution *in-vivo* scans and high resolution histology images. Lowering the resolution can lead to loss of this information and defeats the purpose of using high-resolution scans in the first place. Moreover, this is not representative of clinical and research workflows where high-resolution blockface scans are used as an intermediate modality between in-vivo scans and histology slides (Puonti et al., 2025; Alegro et al., 2016). *Second*, the MP2RAGE sequences in the dataset are both qualitatively and quantitatively different compared to the MP-RAGE sequences seen in the OASIS or LUMIR datasets. This constitutes a significant source of domain shift that leads to poor performance of SLANT on the Ultracortex dataset, making it unsuitable for robust evaluation. This aspect is not discussed and possibly unaccounted for in the original evaluation. We examine the volumes and histograms of the subjects and show that the MP-RAGE sequences (corresponding to subjects `sub-37, sub-45, sub-57`) indeed look qualitatively different than the MP2RAGE sequences in Figure 5. Specifically, histograms of the MP2RAGE sequences are characterized by two or three peaks, close to the extreme values of the intensity range, while the MP-RAGE sequences have a more unimodal distribution with a single dominant peak.

**Evaluation**. To account for the possible effect of domain shift on the performance of SLANT, we perform an alternative evaluation that leverages the high-quality manual segmentations already provided as part of the dataset. MP2RAGE sequences in the dataset provide excellent gray/white matter contrast, making it a practical testbed for evaluating performance of registration algorithms. **Resolution:** First, we affinely register all images to the `sub-3` subject's MP2RAGE image. This brings all images to a 0.6mm isotropic resolution. Initially, we proposed evaluation of both the Greedy and SyN modes in FireANTs, and VFA on the 0.6mm isotropic resolution images. Unfortunately, VFA runs out of memory for 0.6mm isotropic registration on a GPU with 48GB of memory, highlighting the limitations of deep learning methods for high-resolution image registration. Therefore, we further resample the dataset and labels to 1mm isotropic resolution, and evaluate the performance of the same methods on the 1mm isotropic resolution images. **Multimodality:** Moreover, in the dataset, 3 out of the 12 subjects have MP-RAGE sequences, while the other 9 subjects have MP2RAGE sequences. Registration of and MP-RAGE to an MP2RAGE sequence constitutes a multimodal task, and we use MIND features for FireANTs for every pair of images that have different modalities. VFA does not support any other feature images other than intensities as input, and the evaluation for VFA remains unchanged. We report the Dice scores on four splits of the dataset - (a) MP-RAGE to MP-RAGE ($n = 6$), (b) MP2RAGE to MP2RAGE ($n = 72$), and (c) MP-RAGE $\leftarrow\rightarrow$ MP2RAGE ($n = 54$), and (d) all subjects ($n = 132$).

**Results**. The results in Table 5 and Figure 6 highlight three key insights. First, MPRAGE to MP2RAGE registration is a significantly harder task than either MPRAGE to MPRAGE or MP2RAGE to MP2RAGE registration, illustrated by about an 18 point drop in Dice score compared to the MP2RAGE-MP2RAGE split. The MP2RAGE images are well poised to register gray and white matter boundaries due to the excellent contrast, reaching an average Dice score of upto 0.86 for Greedy version of FireANTs. Second, high-

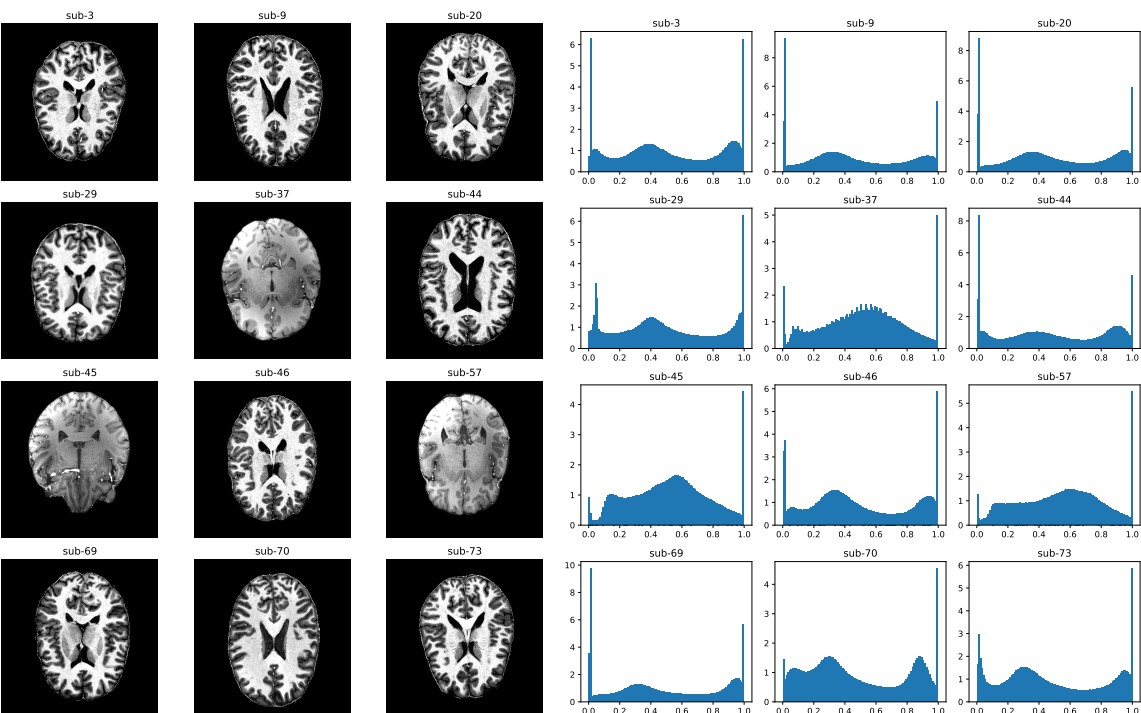

Figure 5: **Multimodal characterization of the Ultracortex dataset. Left** shows axial slices of subjects from the Ultracortex dataset. Out of 12 subjects with labeled segmentations, 3 subjects have MP-RAGE sequence data, and 9 subjects have MP2RAGE sequence data. **Right** shows histograms of the intensity values of the subjects. The MP2RAGE sequences are characterized by two or three peaks close to the extreme values of the intensity range, while the MP-RAGE sequences have a more unimodal distribution with a single dominant peak. The qualitative differences in both the intensity values and histograms are indicative of the multimodal nature of the dataset.

Table 5: Registration performance on Ultracortex dataset across different split types and methods

| Split | Greedy (0.6mm) | SyN (0.6mm) | Greedy (1mm) | SyN (1mm) | VFA (1mm) |
|---|---|---|---|---|---|
| All splits | $0.784 \pm 0.096$ | $0.794 \pm 0.073$ | $0.768 \pm 0.080$ | $0.769 \pm 0.071$ | $0.633 \pm 0.100$ |
| MPRAGE to MPRAGE | $0.804 \pm 0.017$ | $0.803 \pm 0.015$ | $0.788 \pm 0.016$ | $0.787 \pm 0.015$ | $0.648 \pm 0.037$ |
| MPRAGE to MP2RAGE | $0.679 \pm 0.060$ | $0.710 \pm 0.026$ | $0.674 \pm 0.014$ | $0.685 \pm 0.015$ | $0.535 \pm 0.065$ |
| MP2RAGE to MP2RAGE | $0.860 \pm 0.012$ | $0.857 \pm 0.010$ | $0.836 \pm 0.010$ | $0.831 \pm 0.009$ | $0.706 \pm 0.051$ |

resolution registration leads to around a 2 point increase in Dice score for both Greedy and SyN versions of FireANTs essentially obtained for 'free' without any additional domain-specific considerations. Third, the results show that beyond the poor generalization of a representative top performing deep learning method on out-of-distribution contrasts, the methods cannot accomodate multimodal images out-of-the-box. Furthermore, these methods do not scale beyond 1mm isotropic resolution, limiting their applicability to the broad range of high-resolution images and the insights provided by advanced high resolution scan-

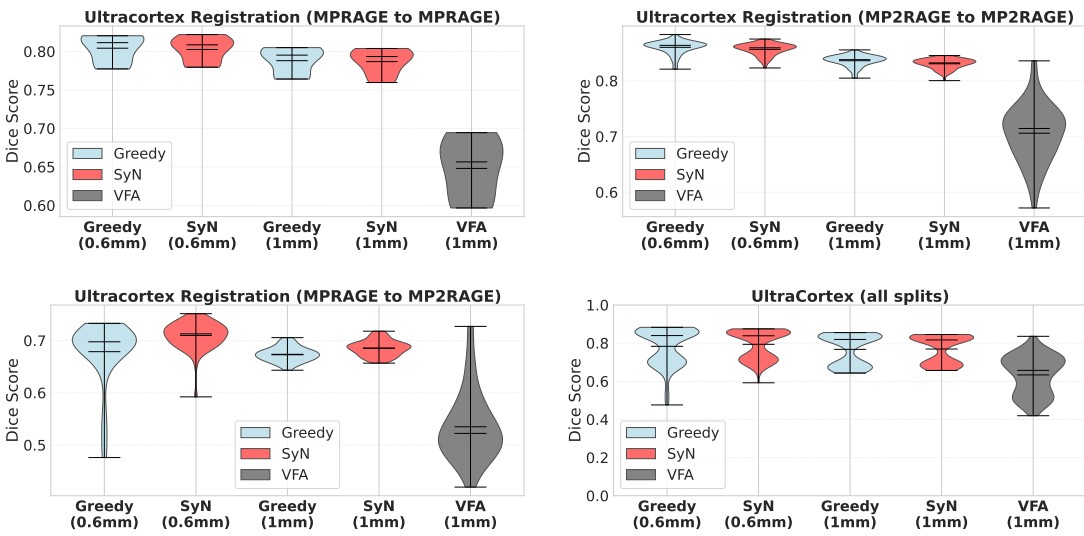

Figure 6: Comparison of the three registration methods on the Ultracortex dataset.

ners, ex-vivo imaging studies, and multimodal integration. With improved efficiency of iterative optimization methods, able to register 0.6mm isotropic images in seconds, they are well positioned to tackle the scale of high resolution imaging workflows pertinent in MRI to histology workflows.

## 7. Deep Learning methods are sensitive to preprocessing choices

An often overlooked limitation of deep learning methods is their sensitivity to preprocessing choices. Most deep learning methods are trained on a fixed set of preprocessing steps, and may perform poorly if the preprocessing steps are not the same as the ones used during training. In contrast to highly controlled evaluation environments like registration challenges, real-world data such as *ex-vivo* hemispheres, histology, and blockface images are rarely standardized to the same preprocessing steps or stereotaxic coordinates. Other domains like MRA imaging can have limited field of view and are highly anisotropic, making it difficult to standardize the preprocessing steps across modalities. Sensitivity to preprocessing choices shifts the burden of preprocessing from the model to the practitioner, who may not be familiar with the preprocessing protocol used during training and might produce suboptimal results.

**Evaluation**. To demonstrate the sensitivity of state-of-the-art deep learning method VFA to preprocessing choices, we perform an ablation study on the NIMH dataset using the SynthSeg segmentation protocol. The NIMH dataset originally contains $208 \times 256 \times 256$ voxels when resampled to 1mm isotropic resolution. Our preliminary experiments with VFA on the original 1mm isotropic T1w images resulted in significantly worse performance than expected. Upon further investigation, we found that VFA performs registration well only if the images are cropped to $192 \times 160 \times 224$ voxels. Therefore, we cropped the images to a smaller region of interest (ROI) of $192 \times 160 \times 224$ voxels and evaluated the performance of

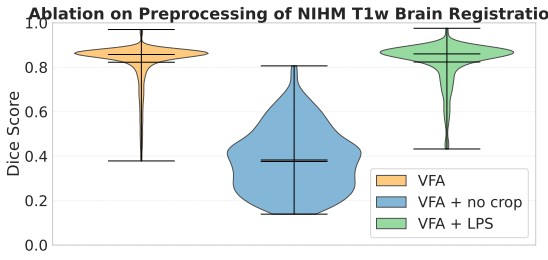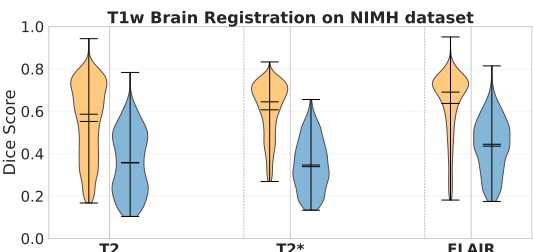

Figure 7: Ablation study on the NIMH dataset showing the effect of preprocessing choices on the performance of the model. **Left** shows the performance of VFA on the cropped images, on the original images (denoted as *no crop*), and on images in the LPS orientation (denoted as *LPS*) on the T1w modality. **Right** shows the performance of VFA on the cropped and original images on the T2w, T2*, and FLAIR modalities.

VFA on these cropped images, upon which we obtained significantly better performance. We also note that VFA was trained on images oriented in RAS frame, and therefore evaluated its performance on the DICOM-standard LPS orientation.

**Results**. Our results in Figure 7 show that VFA performs significantly better on the cropped images across all modalities, and that the performance is significantly worse on the original images, implying that the model is 'locked in' to a particular voxel size. This poses a practical limitation wherein the practitioner may not be able to use the model if the anatomy of interest does not fit inside the field of view of the cropped image. In contrast, iterative methods suffer from no such limitation, and can be readily used by the practitioner without worrying about esoteric preprocessing choices. Fortunately, there is no significant difference in performance by changing the orientation of the images, demonstrating some task understanding and generalization by the model.

## 8. Discussion

**Modern Deep Learning designs show improved Task Understanding on Familiar Modalities** Deformable Image Registration is a highly spatially non local task that requires both global and local context of both the fixed and moving images. Since the dominant formulation of the task is an inverse variational optimization problem, most dominant approaches optimize the warp field directly using various parameteric and non-parametric optimization methods. Early deep learning methods for registration used standard convolutional designs to pose the inverse problem into that of a prediction task. However, these methods suffered from poor generalization to out-of-distribution data, and were unable to register images with different modalities, resolutions, or anatomies. Modern methods borrow design elements from iterative optimization methods (Jian et al., 2024, 2025) to improve generalization to out-of-distribution data. These designs have been shown to be highly effective at improving generalization to domain shift on in-distribution contrasts, and even generalize to human-adjacent species like the Macaque brain, showing that these designs learn task-aware representations. However, these methods still slightly underperform iterative optimization methods on in-distribution data while consuming an order of

magnitude more computational resources at inference time (Jena et al., 2024a). This positions iterative methods as significantly more resource efficient, while still being able to achieve competitive performance, making it a suitable choice for practical applications and deployment on edge devices, and highlighting that there is still room for improvement in efficient designs of deep learning methods for registration.

**Deep Learning methods do not generalize to out-of-distribution data** Contrary to the claims made in the LUMIR challenge, and in accordance with the well-established literature on domain shift, deep learning methods do not generalize to out-of-distribution data despite robust performance on in-distribution data. A segmentation and parcellation algorithm like SLANT generates ROIs that are not always well-defined in terms of intensity boundaries, and are often biased by the internal representation of the model, which could lead to spurious results for intensity-based registration algorithms. To alleviate this potential pitfall, we use SynthSeg to generate high-quality labelmaps for the T2, T2*, and FLAIR modalities on the NIMH dataset, which generates labelmaps whose fidelity is derived from the image itself. Our evaluation shows that the performance gap between optimization and deep learning methods is significantly higher than that on the T1 modality, showing that out of distribution generalization still remains a challenge for deep learning methods. Our data preprocessing and evaluation protocol is made publicly available in our code repository for reproducibility and transparency. Since these results differ markedly from the claims made in the LUMIR challenge, we advocate for evaluation protocols that reflect practical clinical and research workflows rather than conditions that may inadvertently favor particular method classes.

**Deep Learning methods are sensitive to preprocessing choices** A key overlooked aspect of registration challenges in general is the choice of preprocessing steps. Registration challenges typically standardize the data into a common orientation, resolution, and voxel sizes to provide a controlled evaluation environment. However, real-world data including histology, blockface images, *ex-vivo* hemispheres are rarely aligned to any stereotaxic coordinates or have standard resolutions. Registration algorithms must therefore be able to handle a wide range of preprocessing steps, including stereotaxic coordinates, orientations and voxel sizes. Our experiments show that a model trained on images with $192 \times 160 \times 224$ voxels fails catastrophically on the NIMH dataset with the original $208 \times 256 \times 256$ voxels, and cropping extra padding significantly improves performance. Other forms of preprocessing might negatively impact performance and might be hard to debug.

**Code availability** All our preprocessing and evaluation scripts are made publicly available in our code repository (https://github.com/rohitrango/lumirage-evals) for reproducibility and transparency.

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

## Appendix A. Supplementary Material

In this section, we provide additional qualitative results on a few failure modes of the VFA method on out-of-distribution contrasts and preprocessing choices.

### A.1. Sensitivity to Preprocessing Choices

We observed that the performance of VFA is sensitive to preprocessing decisions, particularly the cropping of images prior to registration. In almost all cases, the uncropped (full field-of-view) T1-weighted images from the NIMH dataset led to poor alignments and substantial registration failures, with labelmaps showing significant misalignment and anatomical distortions. Conversely, cropping the images to the same size as the LUMIR challenge dataset mitigated these issues and resulted in noticeably improved registration outcomes. Cropping the images to a fixed size (i.e., $192 \times 160 \times 224$ voxels) may not be a viable option in practical scenarios if the FOV is too tight, or the image is large (e.g. 0.8mm isotropic). In most practical scenarios including clinical ones, registration algorithms are expected to be able to handle a wide range of image sizes and FOVs (including fixed and moving images of different voxel sizes). Figure 8 to Figure 22 illustrate the failure modes for uncropped T1 images.

### A.2. Sensitivity to OOD Image Contrasts

We observed that the performance of VFA is sensitive to the image contrast (i.e. non T1w contrast images). Figure 23 to Figure 35 illustrate the failure modes for T2, T2*, and FLAIR images.

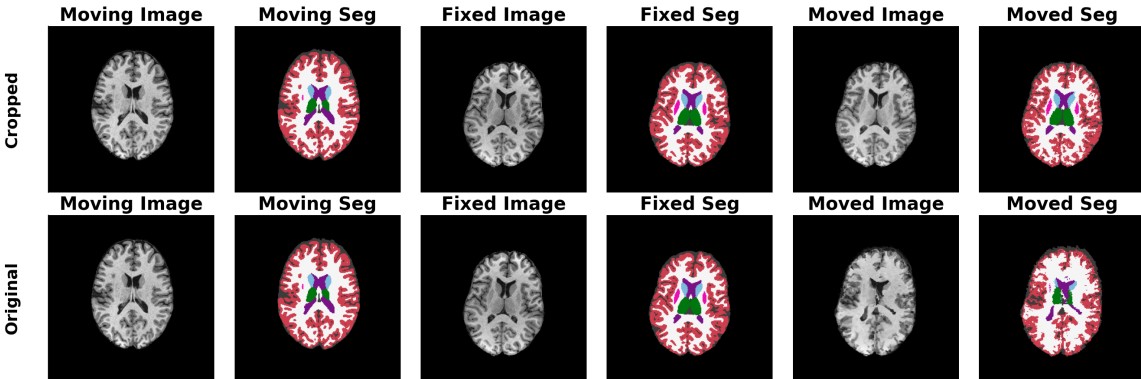

Figure 8: Performance of VFA on the images that are cropped to conform to the LUMIR dataset spec (top row), and the same image pair from the original NIMH dataset (with $256^3$ voxels, bottom). Segmentation labels are shown with FreeSurfer Color LUT.

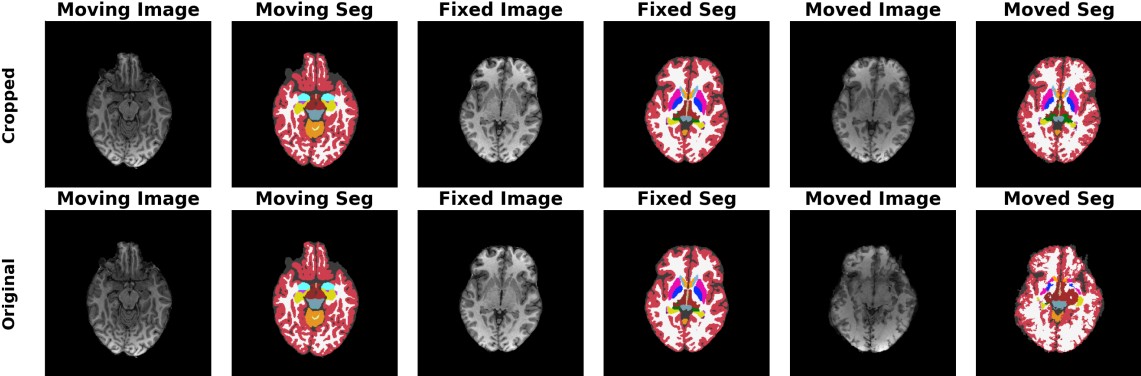

Figure 9: Performance of VFA on the images that are cropped to conform to the LUMIR dataset spec (top row), and the same image pair from the original NIMH dataset (with $256^3$ voxels, bottom). Segmentation labels are shown with FreeSurfer Color LUT.

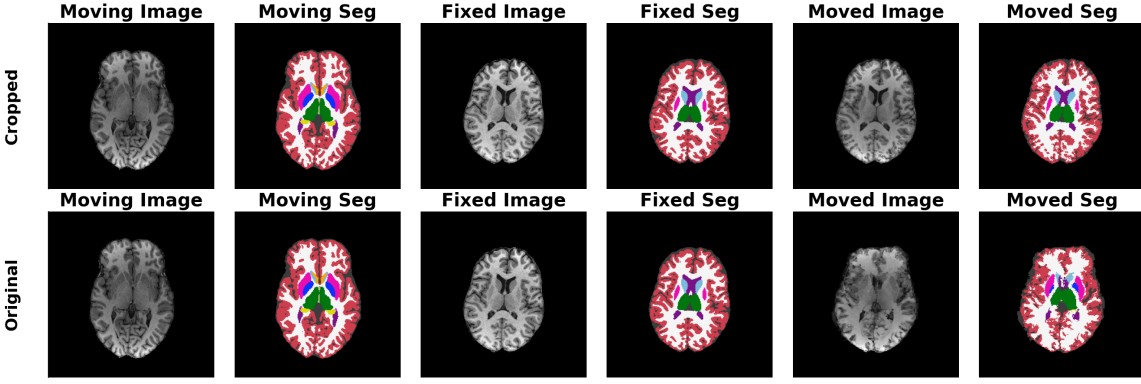

Figure 10: Performance of VFA on the images that are cropped to conform to the LUMIR dataset spec (top row), and the same image pair from the original NIMH dataset (with $256^3$ voxels, bottom). Segmentation labels are shown with FreeSurfer Color LUT.

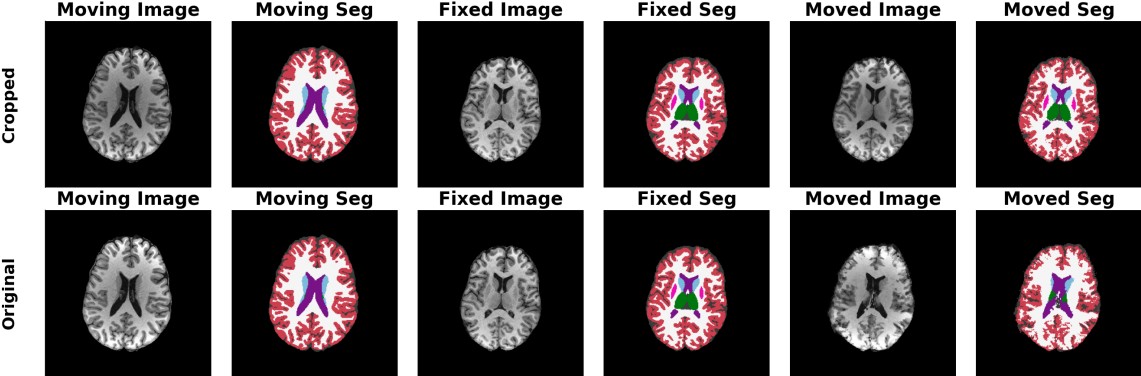

Figure 11: Performance of VFA on the images that are cropped to conform to the LUMIR dataset spec (top row), and the same image pair from the original NIMH dataset (with $256^3$ voxels, bottom). Segmentation labels are shown with FreeSurfer Color LUT.

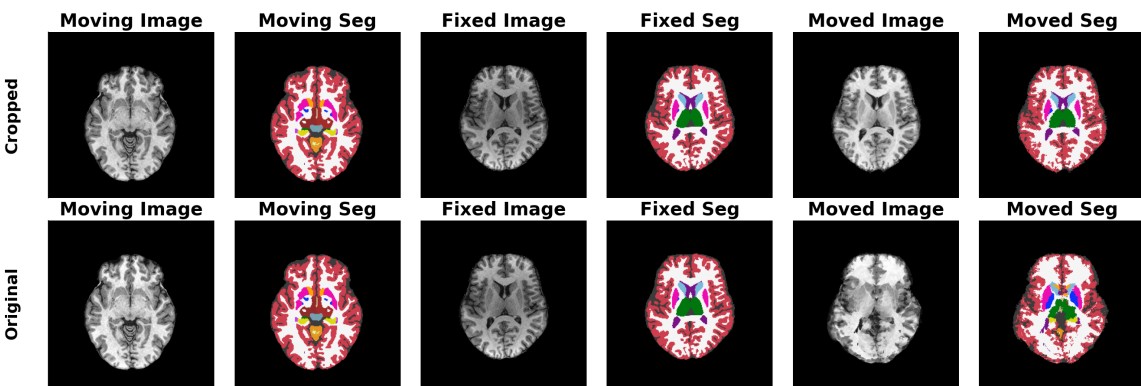

Figure 12: Performance of VFA on the images that are cropped to conform to the LUMIR dataset spec (top row), and the same image pair from the original NIMH dataset (with $256^3$ voxels, bottom). Segmentation labels are shown with FreeSurfer Color LUT.

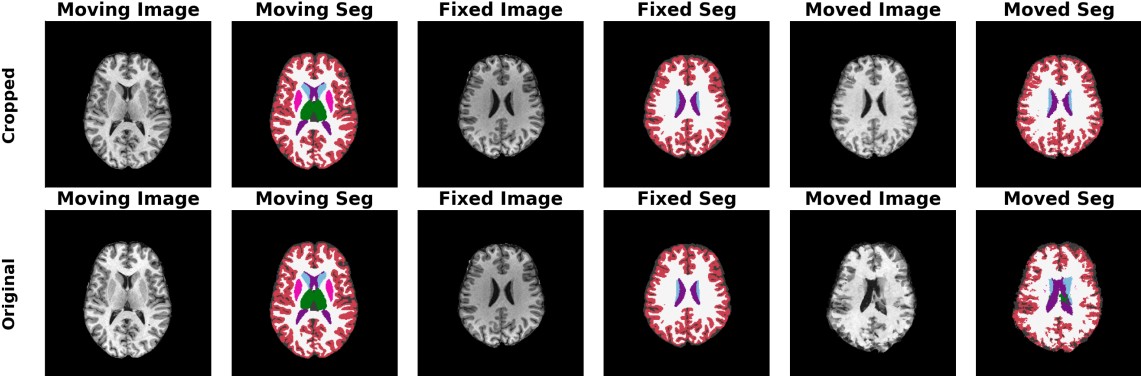

Figure 13: Performance of VFA on the images that are cropped to conform to the LUMIR dataset spec (top row), and the same image pair from the original NIMH dataset (with $256^3$ voxels, bottom). Segmentation labels are shown with FreeSurfer Color LUT.

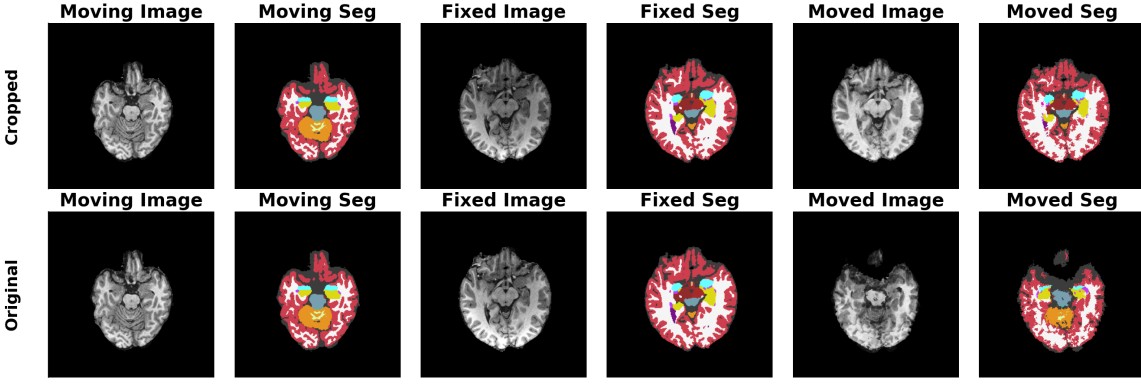

Figure 14: Performance of VFA on the images that are cropped to conform to the LUMIR dataset spec (top row), and the same image pair from the original NIMH dataset (with $256^3$ voxels, bottom). Segmentation labels are shown with FreeSurfer Color LUT.

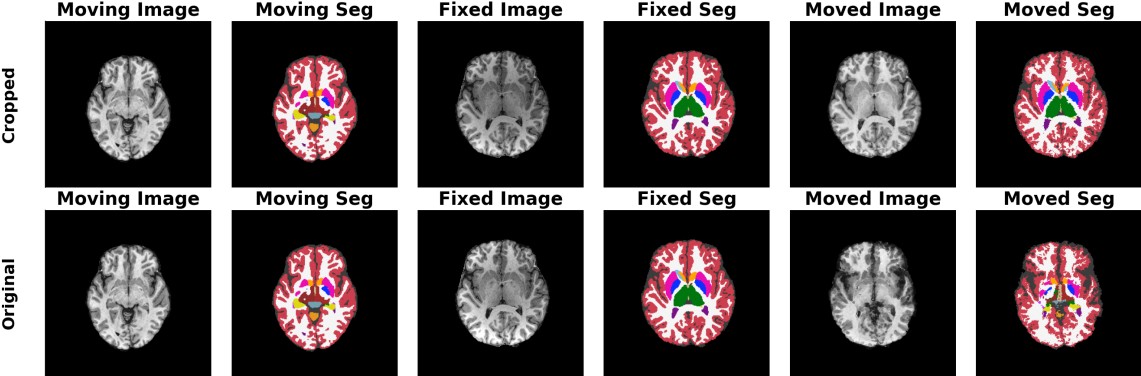

Figure 15: Performance of VFA on the images that are cropped to conform to the LUMIR dataset spec (top row), and the same image pair from the original NIMH dataset (with $256^3$ voxels, bottom). Segmentation labels are shown with FreeSurfer Color LUT.

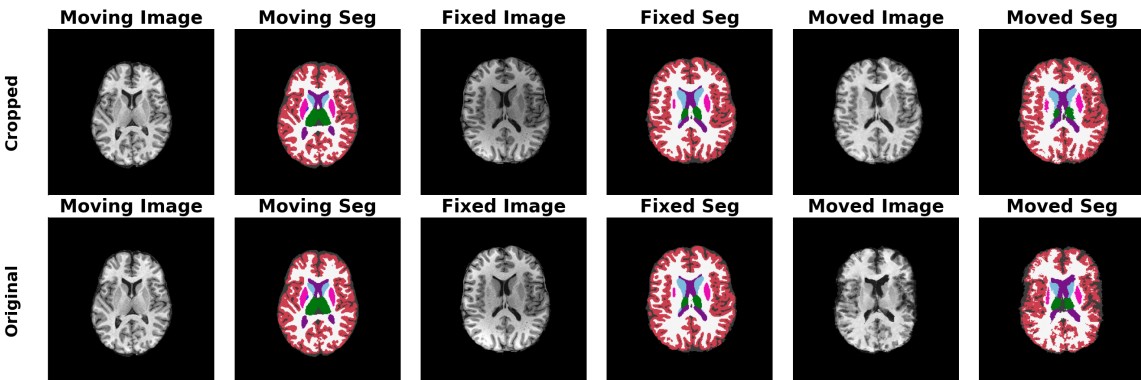

Figure 16: Performance of VFA on the images that are cropped to conform to the LUMIR dataset spec (top row), and the same image pair from the original NIMH dataset (with $256^3$ voxels, bottom). Segmentation labels are shown with FreeSurfer Color LUT.

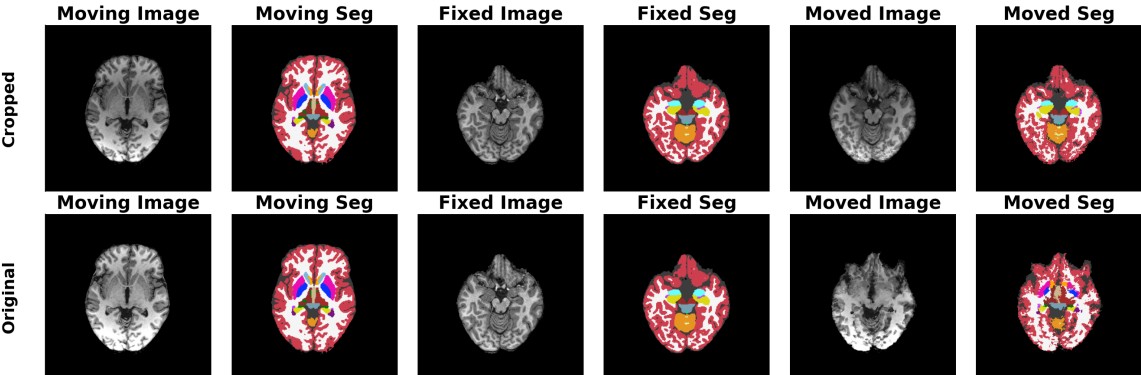

Figure 17: Performance of VFA on the images that are cropped to conform to the LUMIR dataset spec (top row), and the same image pair from the original NIMH dataset (with $256^3$ voxels, bottom). Segmentation labels are shown with FreeSurfer Color LUT.

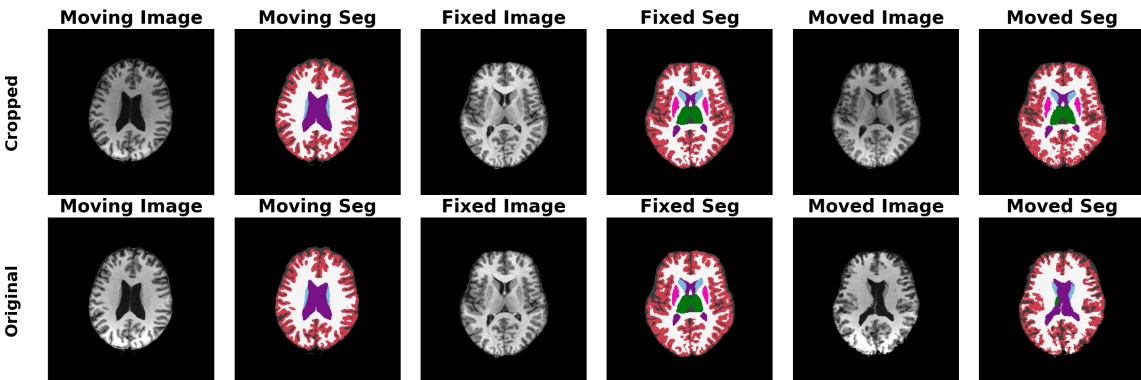

Figure 18: Performance of VFA on the images that are cropped to conform to the LUMIR dataset spec (top row), and the same image pair from the original NIMH dataset (with $256^3$ voxels, bottom). Segmentation labels are shown with FreeSurfer Color LUT.

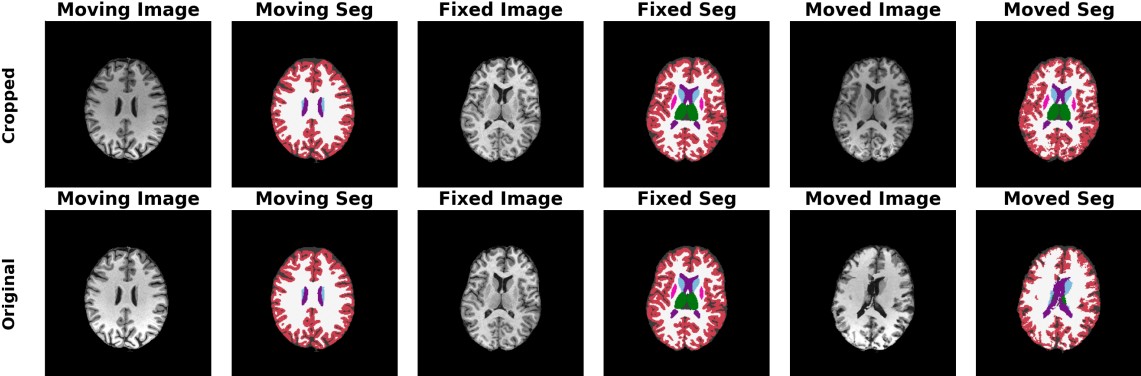

Figure 19: Performance of VFA on the images that are cropped to conform to the LUMIR dataset spec (top row), and the same image pair from the original NIMH dataset (with $256^3$ voxels, bottom). Segmentation labels are shown with FreeSurfer Color LUT.

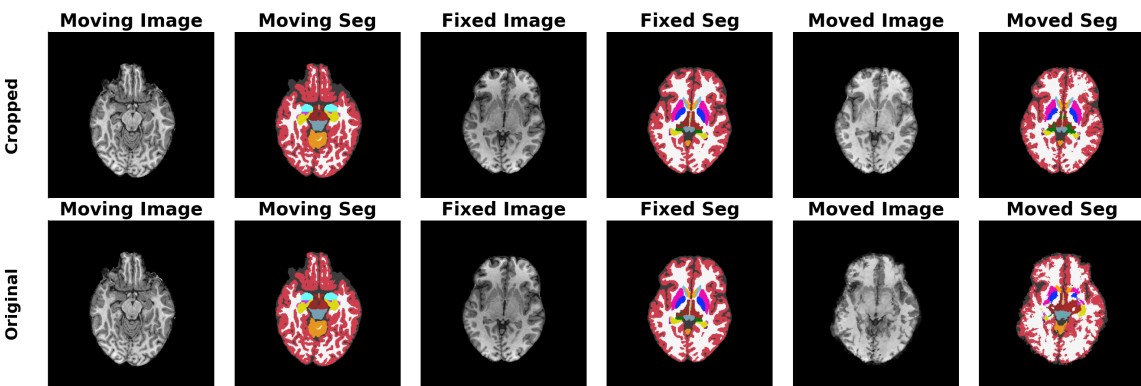

Figure 20: Performance of VFA on the images that are cropped to conform to the LUMIR dataset spec (top row), and the same image pair from the original NIMH dataset (with $256^3$ voxels, bottom). Segmentation labels are shown with FreeSurfer Color LUT.

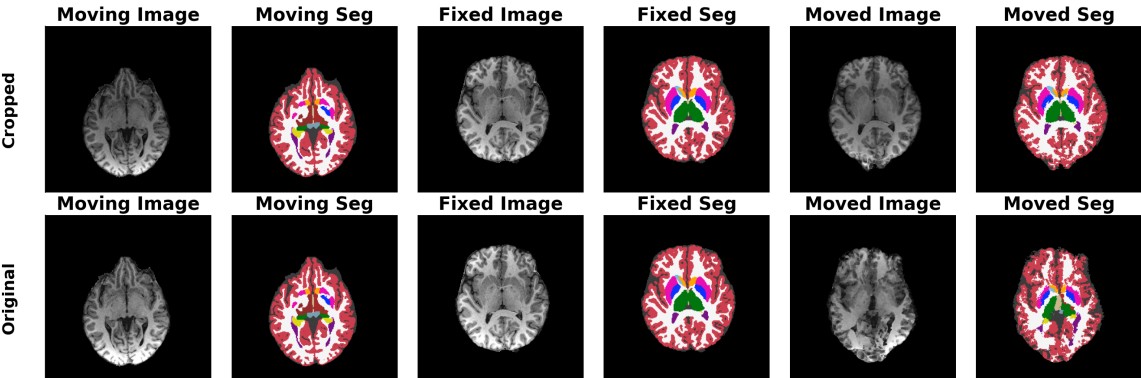

Figure 21: Performance of VFA on the images that are cropped to conform to the LUMIR dataset spec (top row), and the same image pair from the original NIMH dataset (with $256^3$ voxels, bottom). Segmentation labels are shown with FreeSurfer Color LUT.

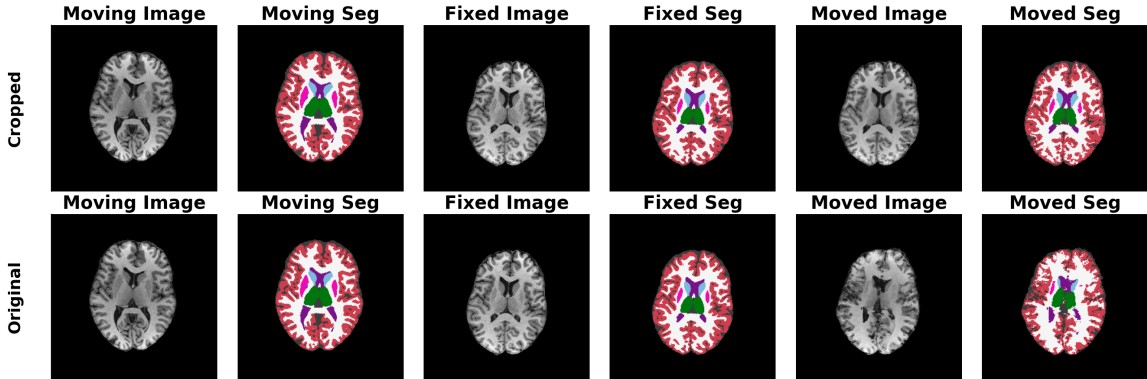

Figure 22: Performance of VFA on the images that are cropped to conform to the LUMIR dataset spec (top row), and the same image pair from the original NIMH dataset (with $256^3$ voxels, bottom). Segmentation labels are shown with FreeSurfer Color LUT.

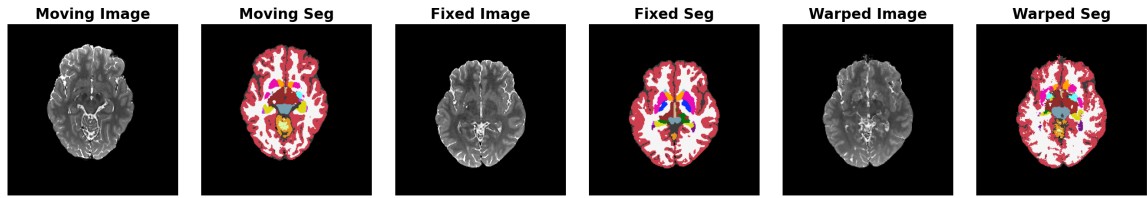

Figure 23: Performance of VFA on T2-weighted images from the NIMH dataset. Segmentation labels are shown with FreeSurfer Color LUT.

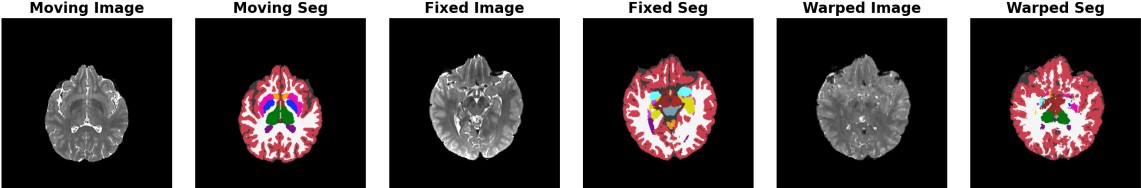

Figure 24: Performance of VFA on T2-weighted images from the NIMH dataset. Segmentation labels are shown with FreeSurfer Color LUT.

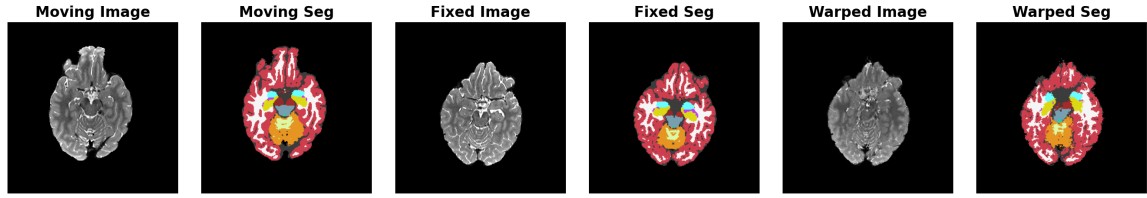

Figure 25: Performance of VFA on T2-weighted images from the NIMH dataset. Segmentation labels are shown with FreeSurfer Color LUT.

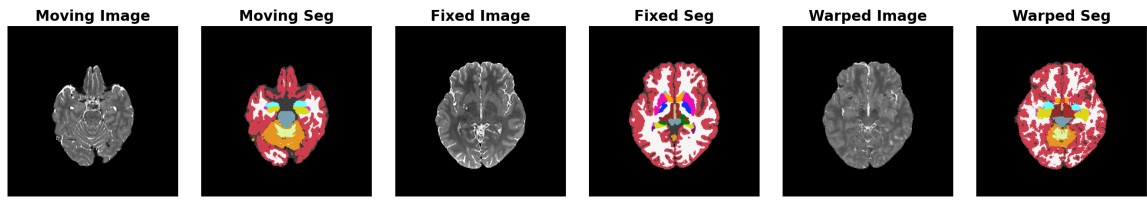

Figure 26: Performance of VFA on T2-weighted images from the NIMH dataset. Segmentation labels are shown with FreeSurfer Color LUT.

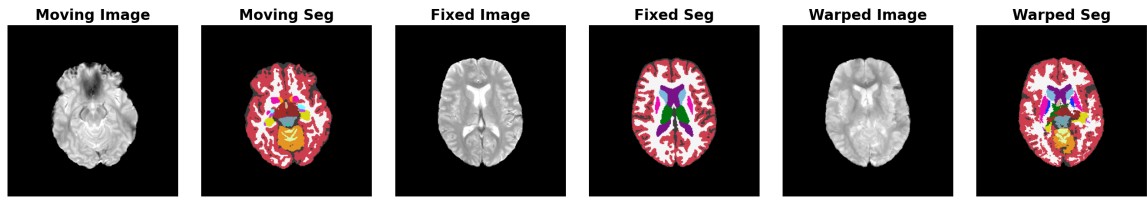

Figure 27: Performance of VFA on T2*-weighted images from the NIMH dataset. Segmentation labels are shown with FreeSurfer Color LUT.

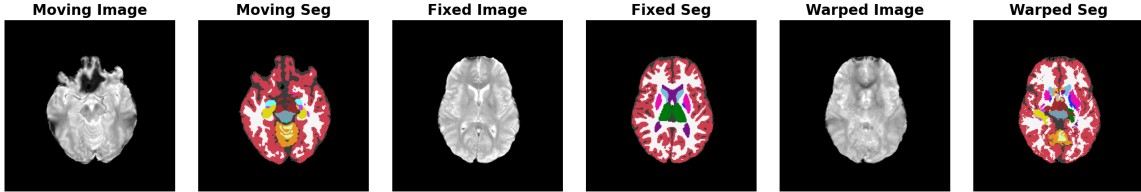

Figure 28: Performance of VFA on T2*-weighted images from the NIMH dataset. Segmentation labels are shown with FreeSurfer Color LUT.

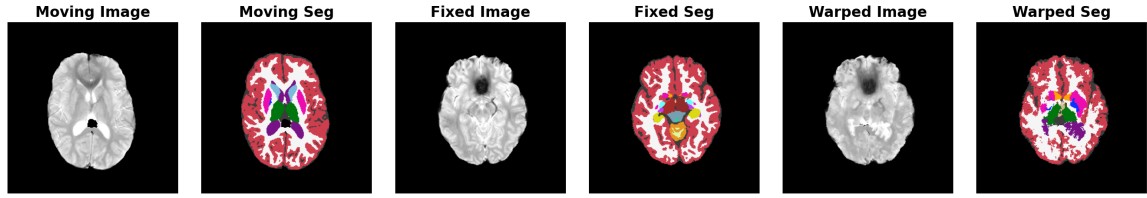

Figure 29: Performance of VFA on T2*-weighted images from the NIMH dataset. Segmentation labels are shown with FreeSurfer Color LUT.

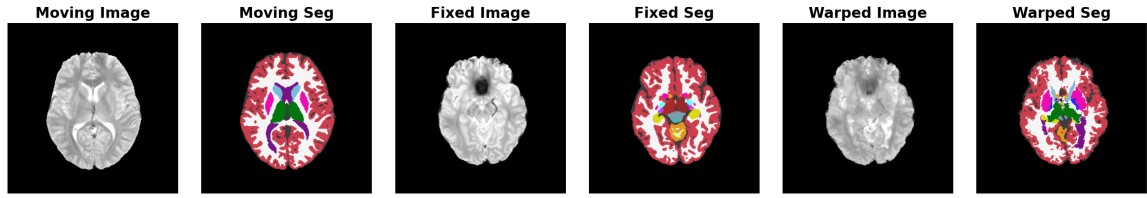

Figure 30: Performance of VFA on T2*-weighted images from the NIMH dataset. Segmentation labels are shown with FreeSurfer Color LUT.

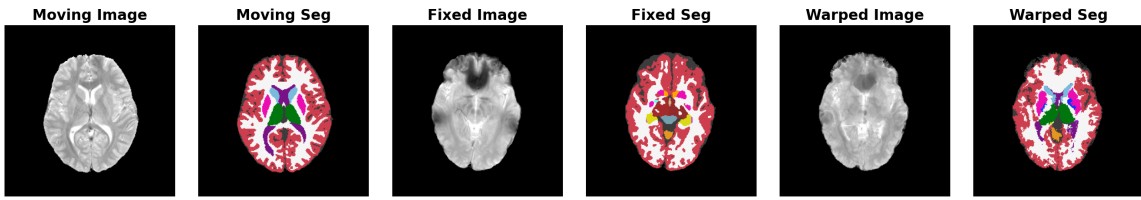

Figure 31: Performance of VFA on T2*-weighted images from the NIMH dataset. Segmentation labels are shown with FreeSurfer Color LUT.

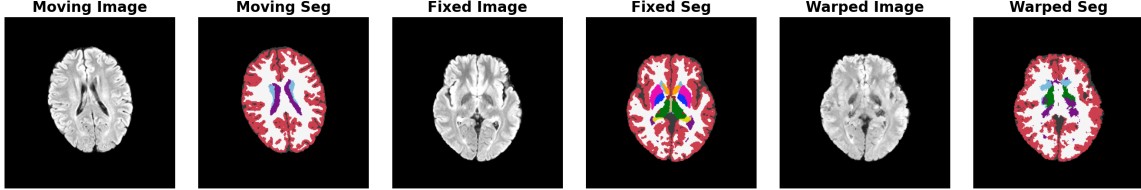

Figure 32: Performance of VFA on FLAIR images from the NIMH dataset. Segmentation labels are shown with FreeSurfer Color LUT.

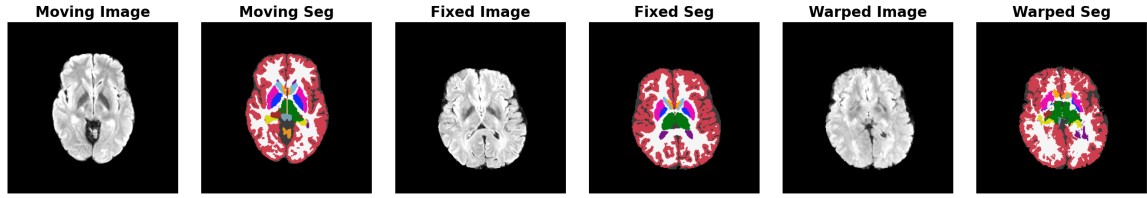

Figure 33: Performance of VFA on FLAIR images from the NIMH dataset. Segmentation labels are shown with FreeSurfer Color LUT.

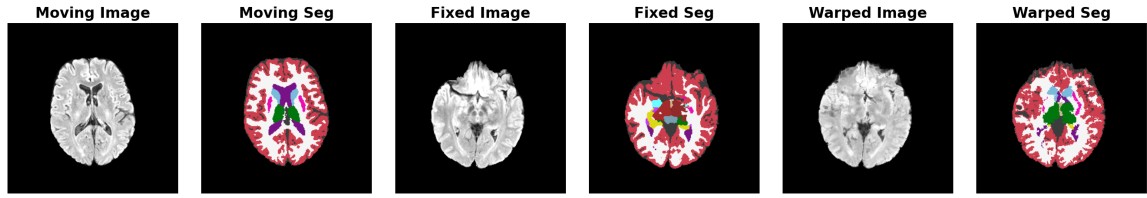

Figure 34: Performance of VFA on FLAIR images from the NIMH dataset. Segmentation labels are shown with FreeSurfer Color LUT.

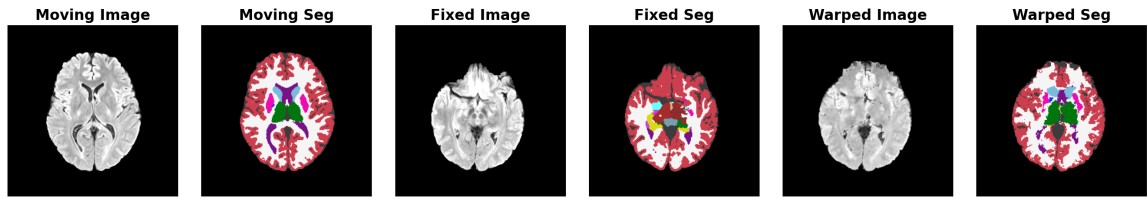

Figure 35: Performance of VFA on FLAIR images from the NIMH dataset. Segmentation labels are shown with FreeSurfer Color LUT.

