# OpenReview forum: "The LU-Mirage - An independent evaluation of the zero-shot claims in the LUMIR challenge"
_MIDL.io/2026/Validation_Papers — MIDL 2026 - Validation Papers Poster_

### Official Review · Reviewer_5TF3 · 2026-01-06

**Confidence:** 4
**Preliminary Rating:** 4
**Final Rating:** 4

**Summary:**

This paper presents an independent and rigorous re-evaluation of the conclusions drawn from the LUMIR challenge. While the LUMIR challenge asserts that modern deep learning registration methods trained on T1-weighted MRI achieve exceptional zero-shot generalization to unseen out-of-distribution contrasts and resolutions, this work challenges those claims. By identifying and mitigating sources of instrumentation bias present in the original evaluation, the authors redesign the experimental protocols. Key contributions include employing more robust label generation strategies, retaining high-resolution data, and conducting sensitivity analyses regarding preprocessing choices.

**Strengths:**

1. This is a "correction" paper. In a landscape often dominated by "leaderboard chasing" and exaggerated claims of generalization, the authors ground their critique in established statistical learning theory.
By questioning counter-intuitive conclusions such as the claim that DL methods outperform iterative optimization on unseen modalities without specific domain adaptation—and refuting them with rigorous experimental data, this work is significant for guiding the community toward more rational and valid evaluation protocols.

2. The authors keenly identify the flaws in the original challenge's protocol, particularly the use of SLANT for evaluating multimodal data. The introduction of SynthSeg, which is robust to out-of-distribution contrasts, provides a much fairer and unbiased benchmark.

**Weaknesses:**

1. The paper primarily compares VFA (as the representative SOTA deep learning method) against FireANTs (Greedy/SyN, representing iterative optimization). While VFA represents modern DL registration, the evaluation would be more robust if it included a diverse range of DL architectures (e.g., classic methods like VoxelMorph or TransMorph) to demonstrate that the observed lack of generalization is a class-wide issue rather than specific to VFA.

2. Regarding the observation that VFA fails to run on 0.6mm isotropic images due to Out-Of-Memory errors, this seems to be more of a technical/engineering constraint rather than a fundamental methodological flaw. Did the authors attempt standard engineering solutions used in deep learning, such as patch-based inference or streaming strategies? Addressing the OOM issue via these techniques to see if DL methods can recover performance at high resolutions would provide a more complete picture.

**Detailed Comments:**

None

**Justification Of Final Rating:**

The authors addressed most of my concerns. Overall, it is a good conference paper, highly suitable for MIDL. It not only challenges existing benchmark results but also educates the readership on how to properly evaluate registration algorithms by avoiding instrumentation bias and ensuring protocols reflect practical clinical workflows.

**Justification Of The Preliminary Rating:**

A good conference paper, highly suitable for MIDL. It not only challenges existing benchmark results but also educates the readership on how to properly evaluate registration algorithms by avoiding instrumentation bias and ensuring protocols reflect practical clinical workflows.

**Questions To Address In The Rebuttal:**

As noted in the weaknesses, could the authors consider expanding the selection of deep learning methods (e.g., including VoxelMorph or TransMorph) to further validate the conclusions?

---

### Official Review · Reviewer_4Ae7 · 2026-01-09

**Confidence:** 5
**Preliminary Rating:** 4
**Final Rating:** 5

**Summary:**

This paper presents an independent and systematic re-evaluation of the zero-shot generalization claims of the LUMIR challenge for deep learning–based deformable image registration. The authors assess performance across multiple datasets, modalities, resolutions, and species, with particular attention to instrumentation bias, preprocessing sensitivity, scalability, and clinical relevance.

**Strengths:**

- Strong alignment with validation track goals.
- The authors evaluate registration performance across multi-aspect evaluation.
- The discussion and experimental control of instrumentation bias.
- Use of effect size (Cohen’s d) alongside descriptive statistics.
- The paper consistently relates findings back to realistic clinical and research workflows.

**Weaknesses:**

- Benchmark citation and contextualization (i.e., VFA) are insufficiently explicit.
- Limited qualitative visualization of failure modes.
- While the authors correctly argue that large sample sizes make p-values less informative and instead emphasize effect sizes, a brief justification is recommended.

**Detailed Comments:**

While the paper reports descriptive statistics and effect sizes, it does not include formal statistical testing to assess robustness across samples. For a validation-focused study, even simple paired tests or confidence intervals on key metrics would strengthen claims of reproducibility and reliability.

In addition, although the authors identify several failure modes (e.g., modality shift and resolution mismatch), very few qualitative examples are provided to illustrate these issues. Including visual registration overlays or difference maps would substantially improve interpretability and translational relevance.

**Justification Of Final Rating:**

Constructive feedback that balances strengths and limitations. Suggestions on adding basic statistical tests and more qualitative visuals are well-justified and would meaningfully strengthen validation impact.

**Justification Of The Preliminary Rating:**

Overall, the paper aligns well with the goals of the Validation Studies Track by emphasizing methodological rigor, independent verification, and translational realism.  The main limitations non-critical.

**Questions To Address In The Rebuttal:**

Please check the weakness and detailed comments section.

---

### Official Review · Reviewer_T1vC · 2026-01-09

**Confidence:** 4
**Preliminary Rating:** 4
**Final Rating:** 5

**Summary:**

The authors challenge the zero-shot performance claims on out-of-distribution datasets made in the LUMIR challenge evaluation by independently re-evaluating modern deep learning registration methods under more rigorous evaluation protocols. They also pay attention to instrumentation bias to ensure a fair comparison.

**Strengths:**

The experiments are well designed. The results can help prevent researchers from being misled by overstated zero-shot generalization claims, and encourage more realistic, bias-aware evaluation practices for benchmarking registration methods.

**Weaknesses:**

In the paper, the authors shows the performance of VFA is significantly worse on the SLANT labelmaps, which is not the case in the LUMIR challenge. The authors assume this is due to preprocessing conditions and label interpolation schemes. But I'm thinking if is this because the implement issue.

**Detailed Comments:**

Some terms are not defined or properly referenced. For example, VFA is not defined in the paper. Please double check the all the terms in the paper.

**Justification Of Final Rating:**

The authors addressed all my concrete concerns with clear sanity checks and reproducible details (code, configs, and preprocessing). The revised paper provides a valuable re-evaluation of LUMIR’s zero-shot claims.

**Justification Of The Preliminary Rating:**

The paper makes a useful, careful independent re-evaluation that can change how people interpret LUMIR’s zero-shot claims. If the authors can verify that the evaluated implementations are correct (e.g., code version, weights, and settings), the paper would be substantially more useful and credible.

**Questions To Address In The Rebuttal:**

1.Since the manuscript does not cite a reference for the VFA method, I can only assume it refers to the work described here: https://arxiv.org/pdf/2407.10209 and the corresponding code repository: https://github.com/yihao6/vfa/. If you used this implementation for the re-evaluation, could there have been any LUMIR-specific modifications that are not reflected in the public release? Did you confirm with the VFA authors about the code used for the LUMIR challenge? I am trying to assess whether VFA was reproduced faithfully.

2.I noticed you reported an out-of-memory issue on a 48 GB GPU. Did you consider running the method on a larger-memory GPU (e.g., a H100 with 96 GB) to enable evaluation at the target resolution? Alternatively, did you explore memory-saving strategies (e.g., checkpointing) to avoid changing the evaluation resolution?

---

### Author Rebuttal · Authors · 2026-01-24

**Rebuttal:**

We appreciate the Reviewers' overall positive feedback and constructive insights, valuable insights, and detailed comments!

In the revised manuscript and supplementary materials (attached), we have made the following updates:
- Supplementary: Added qualitative visualizations of failure modes
- Tables: added statistical significance results (permutation tests)
- Expanded discussion based on results of statististical testing
- Added missing information about VFA, and briefly discussed choice of baselines
- Fixed typos

We have also attached a "Highlighted changes" version of the manuscript (named `highlighted-changes.pdf`) with blue underlined content representing additions, and red strikethroughs representing deletions.

Individual reviewer comments are addressed in the sections below. We thank the reviewers once again for their time.

**Supporting Material:**

/attachment/807fb12d8e3c41bd3e939c22ad7b11c6e315e870.zip

---

### Meta-Review · Area_Chair_LPBy · 2026-02-06

**Recommendation:** Accept (Poster)
**Confidence:** 4

**Metareview:**

This paper provides an independent re-evaluation of the “zero-shot” generalization claims made in the LUMIR registration challenge. Reviewers agreed it is well aligned with the Validation Track: the work is careful about instrumentation bias, uses more robust labeling/evaluation choices, and tests across contrasts, resolutions, and settings that better reflect realistic workflows. The main takeaway is clear and useful -- deep learning methods can be competitive in-distribution, but do not reliably generalize across OOD contrasts/resolutions “for free,” and results can be highly sensitive to preprocessing.

The main concerns in review were whether key baselines (in particular VFA) were reproduced faithfully, whether the paper provided enough context/citations for baselines, and whether the quantitative claims would benefit from clearer statistical testing and more qualitative failure examples. The rebuttal addressed these points well: the authors clarified the VFA setup and performed sanity checks, added missing baseline descriptions, and strengthened the empirical evidence with additional qualitative visualizations and statistical significance analyses (including permutation-style testing to avoid over-interpreting tiny p-values with large n).

Overall, this is a strong validation paper that improves how the community should interpret LUMIR’s headline claims and sets a higher bar for fair, transparent evaluation protocols. I recommend acceptance.

---

### Decision · Program_Chairs · 2026-02-14

Accept (Poster)